# DiODe v2: Unambiguous and Fully-Automated Detection of Directional DBS Lead Orientation

**DOI:** 10.3390/brainsci11111450

**Published:** 2021-10-31

**Authors:** Till A. Dembek, Alexandra Hellerbach, Hannah Jergas, Markus Eichner, Jochen Wirths, Haidar Salimi Dafsari, Michael T. Barbe, Stefan Hunsche, Veerle Visser-Vandewalle, Harald Treuer

**Affiliations:** 1Department of Neurology, Faculty of Medicine, University of Cologne, D-50937 Cologne, Germany; hannah.jergas@uk-koeln.de (H.J.); haidar.dafsari@uk-koeln.de (H.S.D.); michael.barbe@uk-koeln.de (M.T.B.); 2Department of Stereotactic and Functional Neurosurgery, Faculty of Medicine, University of Cologne, D-50937 Cologne, Germany; alexandra.hellerbach@uk-koeln.de (A.H.); markus.eichner@uk-koeln.de (M.E.); jochen.wirths@uk-koeln.de (J.W.); stefan.hunsche@uk-koeln.de (S.H.); veerle.visser-vandewalle@uk-koeln.de (V.V.-V.); harald.treuer@uk-koeln.de (H.T.)

**Keywords:** deep brain stimulation (DBS), directional stimulation, directional electrodes

## Abstract

Directional deep brain stimulation (DBS) leads are now widely used, but the orientation of directional leads needs to be taken into account when relating DBS to neuroanatomy. Methods that can reliably and unambiguously determine the orientation of directional DBS leads are needed. In this study, we provide an enhanced algorithm that determines the orientation of directional DBS leads from postoperative CT scans. To resolve the ambiguity of symmetric CT artifacts, which in the past, limited the orientation detection to two possible solutions, we retrospectively evaluated four different methods in 150 Cartesia™ directional leads, for which the true solution was known from additional X-ray images. The method based on shifts of the center of mass (COM) of the directional marker compared to its expected geometric center correctly resolved the ambiguity in 100% of cases. In conclusion, the DiODe v2 algorithm provides an open-source, fully automated solution for determining the orientation of directional DBS leads.

## 1. Introduction

Directional leads have become the new standard in deep brain stimulation (DBS) for Parkinson’s disease and Essential tremor. Their ability to direct stimulation perpendicular to the lead may reduce therapeutic amplitudes [1] and increase side effect thresholds [2]. However, directional DBS leads vastly increase the number of possible stimulation parameters, of which an optimal subset needs to be chosen in an individual patient. This selection process can be guided by a priori neuroanatomical knowledge [3], probabilistic sweet spots [4,5] or individual neuroimaging [6], but detailed information about the lead’s location and orientation with respect to the surrounding anatomy is needed. Previous results showed that the precise placement of a directional lead into the desired orientation is difficult due to lead torsion [7], which additionally can lead to changes in a lead’s orientation within the first days after implantation [8,9,10]. Consecutively, lead orientation needs to be derived from postoperative imaging, and different methods have been proposed for different imaging techniques [11,12,13]. We previously demonstrated that lead orientation can be reliably detected using the open-source Directional Orientation Detection (DiODe) algorithm, which analyzes the streak artifacts generated by different elements of the directional lead on postoperative CT scans [14,15]. While being highly accurate, the algorithm may need some manual intervention by selecting the CT slices on which the artifacts are most visible. Furthermore, due to the 180° symmetry of the artifacts, DiODe in the past offered two possible, inverse solutions for the orientation of the DBS leadwithout being able to distinguish between them. Recently, Kurtev-Rittstieg et al. suggested two approaches to solve this ambiguity [16]. We now provide version 2 of our DiODe algorithm, which provides (a) a fully automated workflow that optimizes the lead location and artifact detection and (b) can resolve the ambiguity of artifact symmetry. We demonstrate the validity of the algorithm in a large retrospective dataset.

## 2. Methods

### 2.1. Ethics

This retrospective analysis was approved by the local ethics board (Vote: 21-1427). All neuroimaging data retrospectively used in this trial had been acquired as part of routine clinical care in our center and were fully anonymized for this analysis.

### 2.2. DiODe v2

DiODe v2 is an open-source algorithm written in Matlab (The Mathworks, MA, USA), which is released either as a standalone version (https://github.com/Till-Dembek/DiODe_Standalone, accessed on 26 October 2021) or within the Lead-DBS toolbox (www.lead-dbs.org, accessed on 26 October 2021) [17]. An IDL (Exelis Visual Information Solutions, Boulder, CO, USA) implementation is also available upon request. The algorithm requires a postoperative CT scan in Nifti-format as well as the location of the Cartesia™ directional lead (Boston Scientific, MA, USA) within the coordinate system of the CT. In case of the Lead-DBS implementation, both are directly supplied by the Lead-DBS-session. A graphical overview of the algorithm is provided in Figure 1 as well as in our previous publications [7,14,15]. In comparison to previous versions, DiODe v2 includes additional steps, which automatically refine the provided location of the lead based on a linear regression model fitted to slice-wise centers of mass and by automatically detecting the CT slices in which the artifacts needed for orientation detection are most visible. Furthermore, as a new feature, it includes four different methods for resolving the ambiguity resulting from the marker artifact symmetry, which we describe below in more detail.

### 2.3. Center of Mass (COM)

This method, suggested by Kurtev-Rittstieg et al., is based on the center of mass (COM) of the hyperdense volume generated by the stereotactic marker. In a phantom study, their implementation of the COM algorithm was able to correctly resolve the orientation ambiguity in >99 % of cases [16]. In short, the CT is resampled perpendicular to the lead axis at the level of the center of the stereotactic marker with a 0.1 mm resolution. After binarizing this resampled slice at a threshold of 2000 Hounsfield units, the COM is calculated. The resulting COM slightly deviates from the geometric center of the stereotactic marker as defined by the lead trajectory. The direction of this deviation is then compared to the two possible solutions calculated from the hypodense streaks of the marker artifact (see DiODe v1 [14,15]). The solution that is closer to the direction of deviation is chosen by the COM method as the true solution.

For this study, additional implementation of the COM method was also investigated: The COMsagittal implementation resamples not a perpendicular, but a sagittal, ‘in line’ slice which runs through the lead trajectory and is oriented in line with the two possible solutions. A slice from − 1.5mm below to +1.5 mm above the marker center, thus covering the whole marker, is used, again resampled at a 0.1-millimeter resolution. The center of mass is then calculated as described above.

### 2.4. Asymmetric Sampling of the Marker (ASM)

This method, again suggested by Kurtev-Rittstieg et al., investigates asymmetries in the intensity profile of the CT artifact generated by the marker. Postulating that intensities within the marker artifact are greater in the direction of the true orientation versus the direction of the inverse orientation, the heights of the two peaks within the intensity profile are compared for both solutions. In Kurtev-Rittstieg et al.’s phantom study, this method successfully resolved the marker ambiguity in >96% of cases [16].

### 2.5. Star Artifact Symmetry (STARS)

This method relies on the expected difference of the hypodense streak artifacts generated at the level of the segmented contacts for the two inverse orientation solutions. In brief, the expected locations of the streak artifacts are calculated for both solutions. By then sampling the intensity profile at the intersections with the expected streak artifacts, a similarity index between the real streak artifacts and the expected artifact locations can be calculated [14]. By comparing differences in the similarity indices of the two solutions, the real solution is chosen.

### 2.6. Dataset

To validate the different methods, we used retrospective data from 81 patients (*n* = 158 leads) who were implanted with Cartesia™ directional DBS leads. Patients had received routine CT scans, in most cases on the first postoperative day. Since we needed to validate the algorithms solution regarding the marker ambiguity, patients were only included if they had also received additional intraoperative X-ray scans, on which the true orientation solution could be verified visually. Importantly, X-ray scans were not used to calculate an exact orientation angle [15], but only for visual confirmation of which of the two possible solutions provided by DiODe was the correct solution. Previous validations of the DiODe algorithm have clearly demonstrated the impact of the polar angle between the lead and the scanner axis [14,15]. As shown before, the CT artifacts needed for analysis with DiODe begin to disappear with polar angles > 40°, with polar angles > 55° having a severe impact on the accuracy of the algorithm (see data supplement in [7]). For this reason, leads with polar angles > 55° were excluded from analysis.

### 2.7. Analysis

Each patient was analyzed within the Lead-DBS toolbox [17]. Postoperative CT scans were coregistered to preoperative MRI [18], and lead locations were automatically detected using the implementation of the PACER algorithm [19]. Then, we used DiODe v2 to detect lead orientation and compared the resulting solutions regarding marker ambiguity to the ground truth visible on the additional X-rays for the four different algorithms COM, COMsagittal, ASM, and STARS.

## 3. Results

A total number of *n* = 158 directional leads were investigated. Of these, *n* = 8 had a polar angle > 55° with respect to the CT scanner axis, making them unsuitable for analysis with DiODe [14]. *n* = 98 of the remaining *n* = 150 DBS leads targeted the subthalamic nucleus, while *n* = 26 targeted the internal part of the globus pallidus and another *n* = 26 targeted the posterior subthalamic area and the ventral intermediate nucleus of the thalamus. Of the *n* = 150 (74 left, 76 right) leads, the COM algorithm correctly resolved the orientation ambiguity in 150/150 (100%) of cases. COMsagittal correctly identified the orientation in all but one case (149/150, 99.3%). The STARS algorithm performed slightly worse, with 138/150 (92%) orientations being resolved correctly. Of the *n* = 12 leads wrongly classified by STARS, *n* = 11 had a polar angle > 30°. AMS performed the worst, with 123/150 leads being classified correctly (82%). Of the *n* = 27 leads wrongly classified by ASM, *n* = 24 had a polar angle > 30°. Observed deviations from the intended anterior orientation ranged from −136.1 to +97.3° with 17/150 (11.3%) deviating by more than 60°, 31/150 (20.7%) deviating by more than 45°, and 56/150 (37.3%) deviating by more than 30° from the orientation intended during implantation (anterior in our center).

## 4. Discussion

We demonstrate that DiODe v2 provides a fully automatic solution to analyzing the orientation of directional leads from postoperative CT scans and is able to resolve the previously existing ambiguity regarding inverse orientation solutions in up to 100 % of cases.

Since the COM method showed no misclassifications in our large dataset, DiODe v2 now bases its solution on resolving the ambiguity solely on this algorithm. However, due to their high accuracy of >90%, results from the COMsagittal and STARS algorithms are also provided within the GUI and mismatches between algorithms are highlighted to encourage the user to review results more closely. While the ASM method was accurate in a phantom study published as a preprint [16], it underperformed other methods in our study, highlighting the importance of validating such algorithms in real-life datasets.

As long as imaging constraints are respected, i.e., a recommended CT resolution of <1 mm and polar angles between the lead and the CT scanner axis < 55° (recommended <40°) [7], DiODe v2 was able to provide correct results in all investigated cases without any user intervention. As with all neuroimaging techniques, we still encourage users to check the results for plausibility, and the software provides additional steps for manual refinement in case users are discontent with the automatically provided results.

Importantly, DiODe has only been phantom-validated for the Cartesia™ directional lead. While in theory, the algorithm should work with other manufacturers, e.g., the St. Jude Medical Infinity™ directional leads (Abbott Laboratories, IL, USA), imaging constraints need to be evaluated even more carefully for these leads due to the smaller size of their stereotactic markers. Also, DiODe so far has only been validated in cases with one DBS lead per hemisphere. When multiple leads are present in one hemisphere, additional artifacts will be generated which might, in theory, interfere with the algorithm’s results.

One limitation of our analysis is that the postoperative CT scans and the intraoperative X-rays used for visual confirmation of the correct solution were not performed at the same time. As shown in an animal study [9], DBS leads may rotate during the first 24 h after implantation if torque is present. However, these changes are not expected to exceed 60° and thus are not likely to have impacted our analysis regarding the 180° ambiguity of the marker artifacts.

Taken together, DiODe v2 is the first open-source software that allows for the reliable detection of directional DBS lead orientation. Since the software only needs two inputs, namely a postoperative CT scan and two coordinates of the directional lead trajectory, the code can be easily integrated into existing DBS software packages.

## 5. Conclusion

DiODe v2 offers an open-source solution to resolving the orientation of Cartesia™ directional DBS leads from postoperative CT scans in a reliable and fully automated way.

## Figures and Tables

**Figure 1 brainsci-11-01450-f001:**
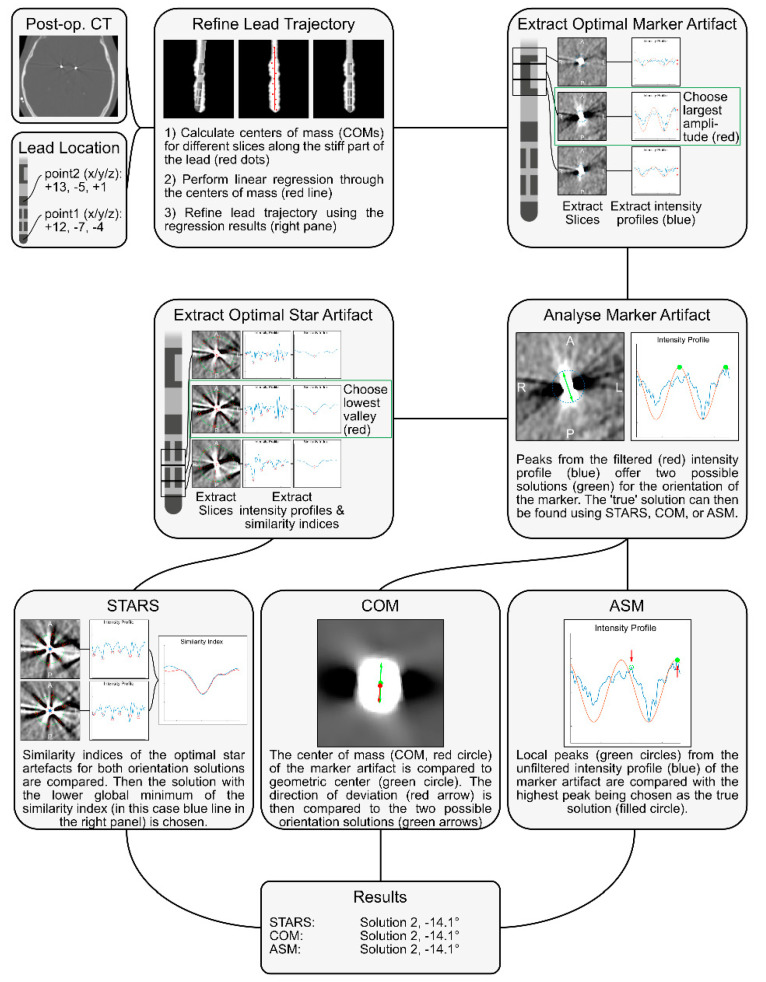
Legend: Flowchart illustrating the different steps of the DiODe v2 algorithm.

## Data Availability

The DiODe v2 algorithm can be accessed as a standalone version (https://github.com/Till-Dembek/DiODe_Standalone, accessed on 8 October 2021) or within the Lead-DBS toolbox (www.lead-dbs.org, accessed on 26 October 2021).

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
