# Peer review of "DiODe v2: Unambiguous and Fully-Automated Detection of Directional DBS Lead Orientation"

_brainsci, 2021, doi:10.3390/brainsci11111450_

Round 1

Reviewer 1 Report

Dembek et al present DiODe v2, an automated algorithm to detect the orientation of directional DBS leads. This is a very useful tool with promising accuracy. A few comments/requests below to help with clarification for the manuscript:

Methods
2.1 & 2.2 What was the time elapsed from DBS lead implantation to CT scan? Did any CT scans include peri-lead artifact such as edema or hemorrhage? Or other neighboring devices such as aneurysm clippings? If so, were these excluded? 

Were the CT scans used in this study limited to 1 DBS lead per brain hemisphere? Has DiODe v2 been tested on images with multiple leads per brain hemisphere (sometimes there is overlapping artifact from neighboring leads in close proximity) 

2.3 Can the authors comment on which resampling method was used on the CT scans? Were they also resampled to a slick thickness of 0.2mm as in Kurtev-Rittstieg et al. 

2.6 Can the authors comment on how the ground truth orientation information from 2D xrays were translated to the 3D CT coordinate system? Or was the xray orientation a visual approximation (as opposed to the specific degrees output from Figure 1)?

Results
Can the authors provide more detail about the one incorrect case via COMsagittal? Was this related to polar angle?

Discussion
line 145 - By "CT resolution", do the authors mean slice thickness? Does DiODe v2 calculate the lead polar angle as part of the toolbox?

Author Response

Reviewer 1:

Dembek et al present DiODe v2, an automated algorithm to detect the orientation of directional DBS leads. This is a very useful tool with promising accuracy. A few comments/requests below to help with clarification for the manuscript:

                We thank Reviewer 1 for their overall assessment of our manuscript!

Methods
2.1 & 2.2 What was the time elapsed from DBS lead implantation to CT scan?

This is an important issue, which we need to address more thoroughly. Our CT-scans are routinely performed on the first day after lead implantation. If there is a suspicion of perioperative complications, CT‑scans might also be performed on the same day as the implantation. Importantly, leads may rotate during the first 24 hours after implantation due to lead torsion (Rau et al. 2021), however, as Rau et al. 2021 demostrated, this orientation change between intraoperative and postoperative imaging did not exceed 60°, even in their animal study where they deliberately applied torsion of up to 360°. Consequetively, changes in orientation cannot be expected to be so large as to confound results regarding the 180 ° symmetry problem addressed in this study.

We added the following to our manuscript:

(Methods, p. 4) Patients had received routine CT-scans, in most cases on the first postoperative day. Since we needed to validate the algorithms solution regarding the marker ambiguity, patients were only included, if they had also received additional intraoperative x‑ray scans, on which the true orientation solution could be verified visually.

(Discussion, p.5) One limitation of our analysis is that the postoperative CT‑scans and the intraoperative x‑rays used for visual confirmation of the correct solution were not performed at the same time. As shown in an animal study[9], DBS leads may rotate during the first 24 hours after implantation if torque is present. However, these changes are not expected to exceed 60° and thus are not likely to have impacted our analysis regarding the 180° ambiguity of the marker artifacts.  

Did any CT scans include peri-lead artifact such as edema or hemorrhage? Or other neighboring devices such as aneurysm clippings? If so, were these excluded? 

Reviewer 1 raises an interesting point which we did not specifically address. Imaging was not screened for possible neighboring abnormalities and so no images were excluded. Regarding edema and hemorrhages, we do not expect these to impact DiODe analyses since the CT‑contrasts generated by the DBS lead and its artifacts have far more extreme houndsfield units (HU) than what would occur during those complications. Even the center of gravity analyses, which in theory might be impacted by other hyperdensities in the immediate vicinity, only takes voxels larger than 2000 HU into account. With an intracerebral hemorrhage not likely to have values larger than 100 HU, an impact on the algorithm is highly unlikely.

Were the CT scans used in this study limited to 1 DBS lead per brain hemisphere? Has DiODe v2 been tested on images with multiple leads per brain hemisphere (sometimes there is overlapping artifact from neighboring leads in close proximity) 

Contrary to the tissue abnormalities discussed above, neighboring leads or other foreign material might indeed introduce artifacts strong enough to theoretically impact the DiODe algorithm. So far, we have only analysed one patient with more than 2 leads (2 VIM/ 2GPi, not part of this dataset), in which the algorithm performed well. We included the following:

(Discussion, p.5) Also, DiODe so far has only been validated in cases with one DBS lead per hemisphere. When multiple leads are present in one hemisphere, additional artifacts will be generated which might, in theory, interfere with the algorithm’s results.

As a side point: Even with only one lead per hemisphere, the contralateral lead already introduces an extra artifact streak which is situated on the line linking the two leads but only to the outside/lateral of each lead. Since these kind of artefacts always occur in the same way, masking and excluding them from analysis is quite straightforward. However, we made the experience that this does not improve DiODe accuracy and thus have not included these steps in the current version of the algorithm.

2.3 Can the authors comment on which resampling method was used on the CT scans? Were they also resampled to a slick thickness of 0.2mm as in Kurtev-Rittstieg et al. 

For the center of gravity methods, the CT is resampled to a 0.1mm resolution – an information now included more prominently in the manuscript:

(Methods, p.3) In short, the CT is resampled perpendicular to the lead axis at the level of the center of the stereotactic marker with a 0.1mm resolution.

(Methods, p.4) The COMsagittal implementation resamples not a perpendicular, but a sagittal, ‘in line’ slice which runs through the lead trajectory and is oriented in line with the two possible solutions. A slice from ‑1.5mm below to +1.5mm above the marker center, thus covering the whole marker, is used, again resampled at a 0.1mm resolution.

2.6 Can the authors comment on how the ground truth orientation information from 2D xrays were translated to the 3D CT coordinate system? Or was the xray orientation a visual approximation (as opposed to the specific degrees output from Figure 1)?

Indeed, for this manuscript, we only used the xrays for visual confirmation of marker orientation regarding the 180 ° ambiguity in the artefacts. While we have published a method which can directly calculate orientation from stereotactic xrays and translate this information into the stereotactic coordinate system (Sitz et al. 2018) for certain stereotactic frames, this method was not used for this study.

We clarified the manuscript accordingly:

(Methods, p.4) Since we needed to validate the algorithms solution regarding the marker ambiguity, patients were only included, if they had also received additional intraoperative x‑ray scans, on which the true orientation solution could be verified visually. Importantly, x-ray scans were not used to calculate an exact orientation angle, but only for visual confirmation of which of the two possible solutions provided by DiODe was the correct solution.

Results
Can the authors provide more detail about the one incorrect case via COMsagittal? Was this related to polar angle?

As suggested by the reviewer, we did investigate this case in more detail. The polar angle was 36.4° and did not impact the analysis. When looking at the results from COMsagittal, the most likely reason for the misclassification was that the calculated geometric center of the lead axis was slightly off so that the center of mass during COMsagittal was situated in the opposite direction from this geometric center. While this was the most rare exception (n=1) it is noteworthy, that the extracted and resampled slice used for COM calculation in COMsagittal also includes the two rings which the manufacturer unfortunately included at the top and bottom of the marker and which, in theory might impact COM calculation – a factor not present in the transversal COM analysis.

Discussion
line 145 - By "CT resolution", do the authors mean slice thickness? Does DiODe v2 calculate the lead polar angle as part of the toolbox?

Indeed slice thickness together with the polar angle are the most important prerequisites for DiODe v2 – albeit CT resolution in the two other dimensions should be below 1mm also. The DiODE v2 GUI provides both – resolution parameters and the calculated polar angle – to the users and highlights suboptimal imaging parameters to inform users in cases results might be impacted.

Reviewer 2 Report

In this manuscript, the authors present an updated algorithm to determine the actual orientation of directional DBS leads with respect to anatomy.  This paper offers solutions to problems identified in their original DiODe algorithm, in which particular CT slices needed to be selected by the user and 2 possible inverse solutions were offered for lead orientation, leading to ambiguity which must be interpreted by the user.  The authors of this paper serve to reconcile these problems by updating the algorithm to enable automatic detection of CT slices in which the streak artifacts are most visible and makes use of four different methods to resolve the ambiguity in lead orientation previously encountered in DiODe version 1.  Accurate knowledge of directional DBS lead orientation can facilitate DBS programming and ultimately lead to better patient outcomes.  It is most appreciated that the authors have created an updated version to DiODe to solve previously identified problems and make the entire process more automated.  I expect this updated algorithm to be highly useful to DBS surgical and clinical management teams.  It is also most appreciated that these algorithms will be available within the Lead-DBS toolbox, which is already highly used within the field.  Overall, this is a clear and well-written paper with many strengths, though would recommend consideration of the following suggestions:

Section 2.2

  • The link for the standalone DiODe v2 (https://github.com/Till-Dembek/Di-54 ODe_Standalone) is not currently live. Therefore, unable to verify the practical use of the algorithm from a user standpoint.  Authors should verify that the link included in the paper is correct and make sure it is active.

Section 2.6 Dataset

  • Would include data on which DBS targets were used. (e.g. were they all STN vs a mix of other surgical targets)
  • As mentioned by the authors in the introduction section, there have been some reports of torsion or shifting of lead orientation in the early post-operative period. Please specify the time range of x-rays used to verify true orientation visually, as this could be pertinent information.

Results

  • If the leads used were from differing surgical targets, would consider adding additional analysis broken down by target to show that the algorithm works consistently independent of surgical target. If a homogenous sampling of only a single surgical target was included then would speak about this in the discussion, as this could affect generalizability of the results and further testing with additional targets may need to be conducted.

Author Response

Reviewer 2:

In this manuscript, the authors present an updated algorithm to determine the actual orientation of directional DBS leads with respect to anatomy.  This paper offers solutions to problems identified in their original DiODe algorithm, in which particular CT slices needed to be selected by the user and 2 possible inverse solutions were offered for lead orientation, leading to ambiguity which must be interpreted by the user.  The authors of this paper serve to reconcile these problems by updating the algorithm to enable automatic detection of CT slices in which the streak artifacts are most visible and makes use of four different methods to resolve the ambiguity in lead orientation previously encountered in DiODe version 1.  Accurate knowledge of directional DBS lead orientation can facilitate DBS programming and ultimately lead to better patient outcomes.  It is most appreciated that the authors have created an updated version to DiODe to solve previously identified problems and make the entire process more automated.  I expect this updated algorithm to be highly useful to DBS surgical and clinical management teams.  It is also most appreciated that these algorithms will be available within the Lead-DBS toolbox, which is already highly used within the field.  Overall, this is a clear and well-written paper with many strengths, though would recommend consideration of the following suggestions:               

We thank Reviewer 2 for this benevolent characterization of our manuscript.

Section 2.2

  • The link for the standalone DiODe v2 (https://github.com/Till-Dembek/Di-54 ODe_Standalone) is not currently live. Therefore, unable to verify the practical use of the algorithm from a user standpoint.  Authors should verify that the link included in the paper is correct and make sure it is active.

This seems to be a problem related to the PDF version made available to reviewers by the submission system (there seems to be the line number “54” intermingled into the link. The real link is:

https://github.com/Till-Dembek/DiODe_Standalone

This link should be active and usable.

Section 2.6 Dataset

  • Would include data on which DBS targets were used. (e.g. were they all STN vs a mix of other surgical targets)

Thank you for raising this important issue. Our dataset included 98 leads targeting the STN, 26 leads targeting the VIM & 26 leads targeting the GPI. As shown in previous publications (Dembek et al. 2019) DiODe is working equally well for these common DBS targets. We included the information in the manuscript:

(Results, p.5) N=98 of the remaining n=150 DBS leads targeted the subthalamic nucleus, while n=26 targeted the internal part of the globus pallidus and another n=62 targeted the posterior subthalamic area and the ventral intermediate nucleus of the thalamus.

  • As mentioned by the authors in the introduction section, there have been some reports of torsion or shifting of lead orientation in the early post-operative period. Please specify the time range of x-rays used to verify true orientation visually, as this could be pertinent information.

This important point was also discussed by Reviewer 1. Indeed there is the possibility for changes in orientation between the intraoperative xray and the postoperative CT scan, performed one day after implantation in most cases (see e.g. Rau et al. 2021). These changes are most likely due to torsion/torque applied to the DBS leads during implantation and fixation. In the Supplementary material of Dembek et al. 2019 we analyzed exact orientations from intraoperative xrays and postoperative CT scans in 90 leads and could show that the maximum observed difference between both imaging modalities and timepoints was about 40°, while it was up to 60° in the animal study performed by Rau et al. where a lot of torsion was applied deliberately. Since for this manuscript, xrays were only used to visually confirm the correct choice of resolving the 180° marker ambiguity, these comparably small differences cannot be expected to impact our results.

We added the following information to our manuscript:

(Methods, p. 4) Patients had received routine CT-scans, in most cases on the first postoperative day. Since we needed to validate the algorithms solution regarding the marker ambiguity, patients were only included, if they had also received additional intraoperative x‑ray scans, on which the true orientation solution could be verified visually.

(Discussion, p.5) One limitation of our analysis is that the postoperative CT‑scans and the intraoperative x‑rays used for visual confirmation of the correct solution were not performed at the same time. As shown in an animal study[9], DBS leads may rotate during the first 24 hours after implantation if torque is present. However, these changes are not expected to exceed 60° and thus are not likely to have impacted our analysis regarding the 180° ambi-guity of the marker artifacts.  

Results

  • If the leads used were from differing surgical targets, would consider adding additional analysis broken down by target to show that the algorithm works consistently independent of surgical target. If a homogenous sampling of only a single surgical target was included then would speak about this in the discussion, as this could affect generalizability of the results and further testing with additional targets may need to be conducted.

As mentioned above, a similar analysis was included in Dembek et al. 2019 so we would like to refrain from adding additional analyses to this manuscript. We did however include the information about different targets in our results.

(Results, p.5) N=98 of the remaining n=150 DBS leads targeted the subthalamic nucleus, while n=26 targeted the internal part of the globus pallidus and another n=62 targeted the posterior subthalamic area and the ventral intermediate nucleus of the thalamus.

Round 2

Reviewer 1 Report

All concerns were appropriately addressed by the authors